# Gender and Age Differences in Performance of Over 70,000 Chinese Finishers in the Half- and Full-Marathon Events

**DOI:** 10.3390/ijerph19137802

**Published:** 2022-06-25

**Authors:** San-Jun Yang, Fan Yang, Yuan Gao, Yan-Feng Su, Wei Sun, Sheng-Wei Jia, Yu Wang, Wing-Kai Lam

**Affiliations:** 1Department of Physical Education and Research, China University of Mining and Technology—Beijing, Beijing 100083, China; 108947@cumtb.edu.cn; 2Li Ning Sports Science Research Center, Li Ning (China) Sports Goods Company Limited, Beijing 101111, China; jiashengwei@li-ning.com.cn; 3School of Physical Education, Yanshan University, Qinhuangdao 066004, China; gaoyuan1107@163.com; 4School of Physical Education and Coaching, Shanghai University of Sport, Shanghai 200438, China; suhui0909@163.com (Y.-F.S.); sunwei1@sus.edu.cn (W.S.); 5School of Kinesiology and Health, Capital University of Physical Education and Sports, Beijing 100091, China; 6Sports Information and External Affairs Centre, Hong Kong Sports Institute, Sha Tin, Hong Kong

**Keywords:** gender, men-to-women ratio, marathon, age, running speed

## Abstract

(1) Background: The aim of the present study was to examine the characteristics of over 70,000 long-distance finishers over the last four years in Chinese half- and full-marathon events; (2) Methods: The available data of all finishers (*n* = 73,485; women, *n* = 17,134; men, *n* = 56,351) who performed half- and full-marathon events in Hangzhou from 2016 to 2019 were further analyzed for the characteristics of gender, age and average running speed; (3) Results: The total men-to-women ratio was the lowest in the half-marathon event (1.86) and the highest in the full-marathon event (17.42). Faster running performance in males than in females and faster average running speed in short-distance runners were shown. Gender and race distance were observed to have the most significant effects on average running speed (*p* < 0.01). For both male and female finishers, the slowest running speed was shown in older age groups (*p* < 0.01) during the full marathon. Our results indicated that the gender difference in performance was attenuated in the longer race distances and older age groups; (4) Conclusions: Understanding the participation and performances across different running distances would provide insights into physiological and biomechanical characteristics for training protocols and sports gear development in different groups.

## 1. Introduction

The health benefits of endurance exercise might partially explain the increase in participation in marathon races during the last decades [1]. In recent years, marathon running has been considered a globally popular physical activity that can cater to the various healthy lifestyle needs of urban residents [2,3]. This running boom has gradually spread around the world. Well-known New York, London, Paris, and Berlin marathon events all had between 30,000 and 40,000 finishers [4]. While distance running used to be a male-dominated sport, today females account for 43% of marathon runners in the USA [5]. Marathon events have developed later in China than in Western countries. The number of marathon events held in China increased from 12 to 53 between 2010 and 2014 [6] to approximately 1100 in 2017 [7], which involved nearly 5 million participants and increased to over 2.2 million participants from 2016 to 2017 [7]. The number further increased to a total of 1900 events in 2019 [1]. The growing popularity of running has inspired a large amount of research on running biomechanics, performance and sport gears in the past decades [8]. Studies have found that proper pace can effectively reduce the risk of musculoskeletal injury [9], and different running strategies should be used in long-distance running according to gender, age and the event the runner is training for [10]. Moreover, differences in running biomechanics between Chinese men and women were observed, with female runners showing greater range of motion in the hip and knee joints, and a smaller shoe-to-ground angle during the heel touch-down phase. This is believed to be a form of self-regulation that women use to reduce the impact of landing, and men rely more on the performance of their shoes to achieve the purpose of buffering [11].

Following the increase in female participation in distance running [12], investigations into gender differences in running mechanics were intensified in the Western world [12,13,14]. The gender difference studies indicated clear differences in females’ body fat and running speed [3], resulting in distinct movement characteristics and injury etiology. Nikolaidis et al. studied the performance and age composition of different genders during marathons, where they found that women achieved their best marathon race time ~5 years earlier in life compared to men. Women’s participation increased disproportionately to men’s participation, leading to an increase in the ratio of men to women [15]. In addition, more and more seniors are joining in marathon races [16]. The sex gap between elite female marathon racers and elite male runners may have reached its limit [17]. The age structure of most male marathon runners is larger and older than that of females (male: 40–44 years; female: 30–34 years) [18].

Although studies have been conducted on anthropometry, physiology and training characteristics have improved our understanding of the predictors of race time [19], as well as age- and gender-related differences in pacing during endurance running [20,21,22,23]. While Western counterparts have been extensively analyzed regarding their running characteristics, little attention has been paid to the Chinese population. Several articles have shown differences in running between Chinese and Western populations; one study found that, compared with Western women, Chinese women use the medial forefoot more during the push-off phase of running [24]. Western female runners have greater ankle valgus angles than males, while there is no significant difference between Chinese females and males [25]. Thus, research findings obtained from Western runners may not be directly applicable to the Chinese population because of racial differences. The present study examined trends of the men-to-women ratio, number of finishers and performances by gender and age groups across four years in half- and full-marathon events, respectively. It was expected that there would be a different men-to-women ratio and different performances across different running events. The comparison could help to better understand potential training requirements for both elite and recreational runners of different age groups, as well as to effectively estimate the demand of running shoes for different gender and age groups. To optimize female performance and health in sport, we need to include women in our analyses in order to better understand peculiarities that may exist in physiology. Therefore, we are happy to enrich the existing pool of knowledge with more data on female participation and performance in marathon racing. Understanding the participation and performance across different running distances (half- and full-marathon events) would provide insights into the physiological and biomechanical characteristics for training protocols for different gender and age groups.

## 2. Materials and Methods

### 2.1. Participants and Data Acquisition

The complete marathon event data for this study were officially obtained from Hangzhou Marathon Organizing Committee (https://www.hzim.org) [26]. The records were collected from all half marathons and full marathons between 2016 and 2019, and were officially certified by the World Athletics Organization. The Hangzhou marathon event included both full and half marathons and runs. Regretfully, the Hangzhou marathon event has been suspended due to the spread of COVID-19 in 2019. The study data included participants who completed the race in the appropriate amount of time. Age and gender information was provided for the period of 4 years. Ultimately, the study included a total of 73,485 participants (male, *n* = 56,351; female, *n* = 17,134).

### 2.2. Procedures

Age intervals of five years were selected to represent age groups among younger and older finishers in their categories. All runners over 71 years old were placed in one category, as there were only a few male runners in the oldest age group, while the oldest male runner was 74 years old. In total, the finishers were classified into 11 age groups; 21–25, 26–30, 31–35, 36–40, 41–45, 45–50, 51–55, 56–60, 61–65, 66–70 and 71+ years. Changes in gender participation are described by the men-to-women ratio (MWR, the quotient of males divided by female completers) [27].

### 2.3. Statistical Analysis

The official race time (i.e., accurate in seconds) was obtained for all finishers in both races. The average running speed in km/h was calculated using the final race time (h) divided by race distance (km) to allow comparison of performances between two long-distance races. All descriptive statistics were reported as mean and standard deviation. Prior to statistical analyses, data distribution normality was verified by visual inspection of histograms and QQ plots [18]. To assess age and gender distribution among finishers in the half- and full-marathon events, a chi-square test (χ^2^) was performed. Statistical differences in marathon performance between 11 age groups and two events were observed. Meanwhile, their interactions were calculated using a two-way ANOVA, post hoc with Bonferroni-corrected tests, and the significance level was controlled at 0.05. All data were organized and summarized using Microsoft Office Excel 2019 (Microsoft Corporation, Redmond, WA, USA) and statistical testing was performed using SPSS 20.0 (IBM, Armonk, NY, USA).

## 3. Results

### 3.1. Participation by Gender, Race Distance, and Age Group

The MWR as well as the total number of male and female finishers in each age group and race distance are presented in Table 1.

The total MWR was 2.15 and 4.99 in the half and full marathon, respectively. A gender × race distance association in participation was shown (χ^2^ = 2294.505, *p* < 0.01, φ = 0.177). A gender × age group association in participation was observed in the half marathon (χ^2^ = 89.091, *p* < 0.01, φ = 0.081) and in the full marathon (χ^2^ = 53.431, *p* < 0.01, φ = 0.050). Furthermore, a race distance × age group association in participation was shown for male finishers (χ^2^ = 1687.054, *p* < 0.01, φ = 0.187) and for female finishers (χ^2^ = 600.388, *p* < 0.01, φ = 0.197) of different age groups. In the half marathon, the lowest MWR was observed in the age group of 26–30 years (1.69), whereas the highest MWR was observed in the age group of 36–40 years (6.12). In the full marathon, the lowest MWR was observed in the youngest age group (4.04), whereas the highest MWR was observed in the age group of 66–70 years (17.41).

### 3.2. Performance (Average Running Speed)

#### 3.2.1. Overall Effects

The two-way ANOVA showed significant effects of gender [F _(1, 73,365)_ = 612.757, *p* < 0.001] and race distance [F _(1, 73,365)_ = 7.914, *p* < 0.005], as well as age group [F _(10, 73,365)_ = 119.550, *p* < 0.001]. Moreover, we found significant interactions for age group × race distance [F _(10, 73,365)_ = 3.763, *p* < 0.001], while no interaction was observed for gender × race distance [F _(1, 73,365)_ = 3.270, *p* = 0.071] and for gender × age group [F _(9, 73,365)_ = 1.329, *p* = 0.216].

#### 3.2.2. Performance by Gender and Race Distance

A significant effect of gender on average running speed is shown (*p* < 0.001) in Figure 1, where male finishers (with the performance of 10.03 ± 1.67 km/h) were faster than female finishers (with the performance of 9.08 ± 1.27 km/h). In addition, a significant effect of race distance on average running speed was observed (*p* < 0.001). Figure 1 also shows that performance in the full-marathon event (9.95 ± 1.71 km/h) was faster than that in the half-marathon event (9.63 ± 1.53 km/h).

No gender × race distance interaction on average running speed was found (*p* > 0.05), while the gender difference was lower in the half-marathon event (+9.98%) than in the full-marathon event (+10.25%). Half-marathon finishers were slower than full-marathon finishers among females (9.02 ± 1.22 versus 9.17 ± 1.35 km/h, respectively, *p* < 0.001), as well as among males (9.92 ± 1.57 versus 10.11 ± 1.73 km/h, respectively, *p* < 0.001).

#### 3.2.3. Performance by Age Group and Race Distance

The age group × race distance interaction had a significant effect on average running speed (*p* < 0.001). Under closer examination of the performance, male finishers had the fastest average running speed of 10.33 ± 1.60 km/h, while female finishers had the slowest average running speed of 8.54 ± 1.03 km/h, regardless of the type of event.

In the half-marathon event, the fastest male age group was 61–65 years (average running speed of 10.32 ± 1.45 km/h), while the slowest male age group was 26–30 years (average running speed of 9.61 ± 1.57 km/h). In the full-marathon event, the male finishers had the fastest speed of 10.39 ± 1.62 km/h in the 46–50 age interval and the slowest speed of 9.16 ± 1.18 km/h in the 71+ age interval (Figure 2a).

In the half-marathon event, female finishers had the fastest average running speed of 9.61 ± 1.21 km/h in the 66–70 age interval, but the slowest average running speed of 8.65 ± 1.13 km/h in the 26–30 age group. In the full-marathon event, the fastest female finisher was observed in the 46–50 age group (9.43 ± 1.28 km/h), while the slowest female was in the 66–70 age group (8.48 ± 0.91 km/h) (Figure 2b).

#### 3.2.4. Performance by Gender and Age Group

Although the main effect of gender × age group interaction on the average running speed (*p* = 0.216) was obtained, Bonferroni post hoc comparisons revealed a significant difference (Table 2).

In both male and female finishers, there was a significant difference between the age group of 21–25 years and the age group of 36–60 years (*p* < 0.01), as well as the age group of 61–55 years (*p* < 0.05). The same difference was found in the age groups of 26–30 years and 31–65 years (*p* < 0.01). 

There was also a difference in the average running speed between age groups of 31–35 years, 36–60 years (*p* < 0.01) and 61–55 years (*p* < 0.05). Another significant difference was observed between the 36–40 age interval and the 41–55 age interval (*p* < 0.01). The performance of the 46–50 age interval showed a significant difference with the 41–45 age interval (*p* < 0.01) and the 51–55 age interval (*p* < 0.01).

## 4. Discussion

The aim of this study was to assess the age, gender and average speed characteristics of over 70,000 long-distance finishers over the last four years.

We found that the 31–35 and 36–40 age groups had the largest number of male finishers in the half-marathon event, while the 26–30 and 36–40 age groups had the largest number of female finishers. These findings are partly in agreement with previous studies, which reported that, regardless of gender, the largest number of finishers was found in the 24–34 age group [28]. In addition, a study conducted in Switzerland found that the two largest numbers of full-marathon finishers were in the age ranges 40–44 and 45–49 in men, but 35–39 and 40–44 in women; this study also reported that most half-marathon participants were 30–34 years old and 45–49 years old in men, but 25–29 years old and 30–34 years old in women [29]. Although there are differences among age groups in Asian and European populations, over 30 years of age appeared to be the dominant age in participants in long-distance running, which may be related to the relationship with physical needs, social influences and disposable time [30]. The study showed that the largest age group participating in full marathon events was usually older than the half-marathon runners. Such results may be due to older runners being more emotionally stable and responsible than younger people [31].

In terms of the number of finishers by race distance, there were ~1.3 times more marathon runners than half-marathon runners, which is not in agreement with the previous findings. The reason for this phenomenon may be that the marathon organizing committee limited the number of half-marathon runners during the registration period. In Switzerland, the total number of half-marathon runners was about 2.6 times higher than the marathon runners between 2000 and 2010 [32]. In Greece, the total number of half-marathon runners was about 3.5 times higher than that in the Oslo Marathon between 2008 and 2018 [28]. This could be explained by the fact that the registered number of half-marathon events is substantially lower than the full-marathon events held in China [33].

It is not difficult to understand the phenomenon that the number of people who finish the half marathon is higher than the number of people who finish the full marathon. The demand for physical fitness in the full marathon is undoubtedly higher. Running long distances can lead to dehydration, damage to muscle tissue and an increase in body temperature, which must be more difficult to overcome during a full marathon [34]. Moreover, the depletion of glycogen stores in the body causes athletes to “hit the wall (HTW)”; the frequency of HTW among elite and non-elite runners was 51%, with the greatest proportion being found in non-elite runners. Thus, HTW also increases the risk of withdrawal as the distance of the competition increases [35].

Our data confirmed that a higher number of male finishers than females was also observed in a longer-distance event, which is similar to the “Marathon des Sables” (7-day competition) with a men-to-women ratio (MWR) of 6.76 [3], the “Western States 100-Mile Endurance Run” (161 km) with an MWR of 5.28 [36], or Double Iron Ultra-Triathlon (MWR: 8.96) to Deca Iron Ultra-Triathlon (MWR: 6.94) [37]. However, over the past few years, an increase in women finishers was observed in half- and full-marathon events, as indicated by the decreased trend of MWR. A plausible explanation for the MWR variation by race distances/intensities might be the consideration of females as relatively “novice” runners compared to males. One study reported that the MWR was decreased from 10.2 in the 1970s to 1.5 in the 2010s in the “New York City Marathon” [38]. Women complete shorter-distance races first and then longer-distance races according to their ability and physiological features. The majority opted for the half marathon, resulting in a higher number of female finishers in the half marathon than in the full marathon. Regarding changes in finishers by gender and age group, similar trends were shown across all race distances, with older groups having a higher MWR than younger age groups.

The current findings showed faster running performance in males than in females and faster average running speed in shorter race distances. This may be attributable to the innate physiological advantage of male runners and gender differences in training habits, with males having greater body weights and lower body fat percentages in terms of physiology [3]. The research pointed out that men and women are born with different muscle fiber properties, and the advantage of men is that the circumference of muscle fibers is larger than that of women, so they are more powerful [39]. This is the difference between men and women in gene expression in human skeletal muscle [40]. Furthermore, male athletes exhibit higher maximal oxygen uptake (VO_2_max) and anaerobic thresholds than females in long-distance running [41]; there is a positive correlation between VO_2_max and thermoregulatory ability [42]. Some articles have pointed out that in the non-competition training phase, men have longer training distances than women and have more weekly training sessions, so the running experience of male runners is higher than that of women [43]; the same goes for training differences for a half marathon [3]. Some researchers have also found that the larger male-to-female ratio in the older group is due to the lower number of female finishers [44]. Our data also found that the average pace of women in half-marathon events increased with age. In previous research, the slowest pace occurred in the older age groups [22]. One of the reasons for this could be that most young female participants in the half marathon were attempting long-distance running for the first time, which makes the pace slower in younger age groups. At the same time, the running speeds of older Chinese women groups were better than Western women. Eastern women are morphologically thinner than Western women [38], thus saving more running economy in long-distance running and are, therefore, faster on average.

There were several limitations in the present study. First of all, the role of environmental conditions, such as detailed data on temperature, humidity, and wind, was not considered. It is reasonable to involve the effect of these environmental parameters on the endurance performance [45]. Secondly, the number of participants was higher than other studies, as four-year data were accessible. Long-term research should be conducted in future. Thirdly, other endurance events such as 10 km races should be included.

## 5. Conclusions

This study found that the number of female finishers in the half- and full-marathon races has increased, but, overall, there are still more males than females. A higher number of younger female finishers participated in both race distances. Moreover, the overall performance of male’s running is better than female’s running, but as the race distance and the age of participants increased, the difference in performance caused by gender gradually weakened. It should be emphasized that the analysis of performance trends is related to changes in MWR based on age group and race distance.

## Figures and Tables

**Figure 1 ijerph-19-07802-f001:**
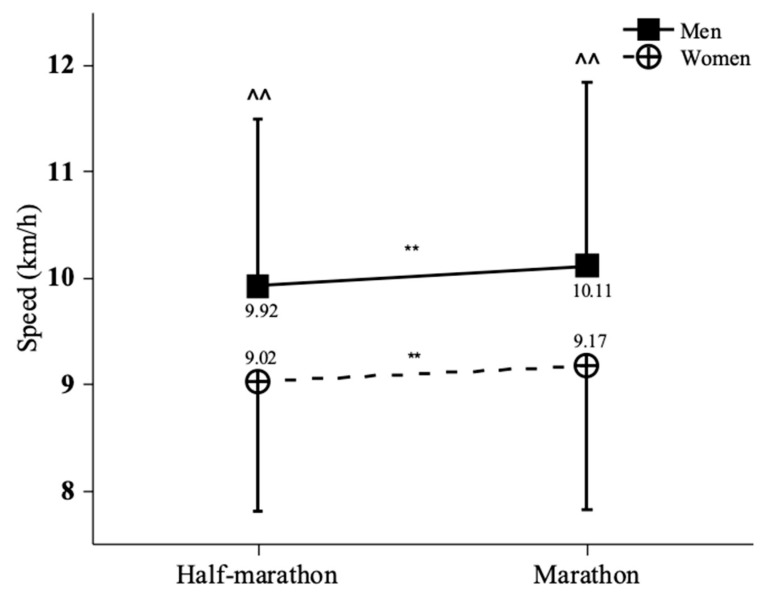
Race speed by race distance and gender. Error bars represent standard deviations. ^^ *p* < 0.001; ** *p* < 0.001.

**Figure 2 ijerph-19-07802-f002:**
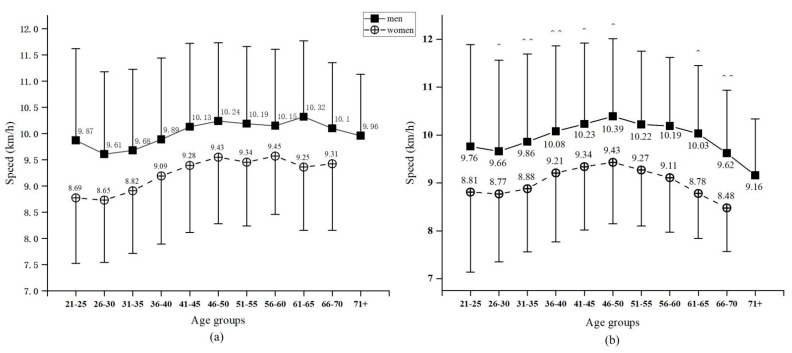
Race speed by age group and gender in the half-marathon event (**a**) and in the full-marathon event (**b**). Error bars represent standard deviations. ^^ *p* < 0.01; ^ *p* < 0.05.

**Table 1 ijerph-19-07802-t001:** Distribution of male and female finishers in each age group and race distance.

	Half-Marathon				Full-Marathon			
Age Groups	Males	Females	Total	MWR	Males	Females	Total	MWR
**21–25**	940	505	1445	1.86	639	158	797	4.04
**26–30**	3667	2166	5833	1.69	3046	714	3760	4.27
**31–35**	4381	2063	6444	2.12	5110	1073	6183	4.76
**36–40**	4360	1712	6072	2.55	6274	1249	7523	5.02
**41–45**	3514	1538	5052	2.28	6733	1325	8058	5.08
**46–50**	2820	1404	4224	2.01	6573	1386	7959	4.74
**51–55**	1428	633	2061	2.26	3569	678	4247	5.26
**56–60**	618	179	797	3.45	1540	193	1733	7.98
**61–65**	208	47	255	4.43	502	64	566	7.84
**66–70**	111	20	131	5.55	209	12	221	17.42
**71+**	28	0	28	-	18	0	18	-
**Total**	22,075	10,267	32,342	2.15	34,213	6852	41,065	4.99

MWR = men-to-women ratio.

**Table 2 ijerph-19-07802-t002:** Bonferroni post hoc tests of age groups.

Age Groups	21–25	26–30	31–35	36–40	41–45	46–50	51–55	56–60	61–65	66–70	71+
**21–25**	-	-	-	****	****	****	****	****	***	-	-
**26–30**	-	-	****	****	****	****	****	****	****	-	-
**31–35**	-	##	-	****	****	****	****	****	***	-	-
**36–40**	##	##	##	-	****	****	****	-	-	-	-
**41–45**	##	##	##	##	-	****	-	-	-	-	-
**46–50**	##	##	##	##	##	-	****	-	-	-	-
**51–55**	##	##	##	##		##	-	-	-	-	-
**56–60**	##	##	##	-	-	-	-	-	-	-	-
**61–65**	#	##	#	-	-	-	-	-	-	-	-
**66–70**	-	-	-	-	-	-	-	-	-	-	-
**71+**	-	-	-	-	-	-	-	-	-	-	-

* represent male, # represent female, ** or ## *p* < 0.01; * or # *p* < 0.05.

## Data Availability

The data presented in this study are available on request from the corresponding author.

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
