# Peer review of "Gender and Age Differences in Performance of Over 70,000 Chinese Finishers in the Half- and Full-Marathon Events"

_ijerph, 2022, doi:10.3390/ijerph19137802_

Round 1
Reviewer 1 Report
This study examines gender and age differences in the performance of Chinese finishers in the half-and full-marathon events. The study is quite intriguing with the appropriate and thorough methods. It is rather well written, with significant findings that can help athletes, scientists, coaches, and other sports practitioners. However, some comments and issues should be addressed, particularly in the study rationale.
I am not a native English speaker; however, I have noticed some typos and other language issues. Please try to improve overall English writing where possible.
Introduction:
1. "The growing popularity in the running has evolved a large amount of research on running biomechanics, performance and sports gears in the past decades." This sentence indicates "a large amount of research" and cites only one. Please add more relevant citations here.
2. Too much was spent on the previous research regarding the age and gender differences in western runners. Only one sentence was a clear rationale for this study: "research findings obtained from western runners may not directly be applicable to the Chinese population because of racial differences", however, this is not enough. At least one additional paragraph is needed to elaborate on these potential racial differences. Otherwise, this manuscript uses the same methodology on different populations, which is not enough for a journal of this quality.
3. It is unclear how understanding the participation and performances would provide insights into the biomechanical characteristics for training protocol and sports gear developments. This manuscript is not analyzing running gait or posture, or similar. Please elaborate or rewrite this part (also in the Abstract).
Discussion:
This comment is related to one presented in the introduction. The rationale for this study is that Chinese runners might differ from western runners. Yet, there is no explanation of the obtained results regarding the potential racial differences. Please elaborate on that in the discussion.
References:
Consider using this novel reference regarding the long-distance race differences:
Cuk, I., Nikolaidis, P. T., Villiger, E., & Knechtle, B. (2021). Pacing in Long-Distance Running: Sex and Age Differences in 10-km Race and Marathon. Medicina, 57(4), 389.
Figures:
Figure 2 has poor visibility. Please improve that.
Reviewer 2 Report
One of the unique aspects in this study is that it is a longitudinal study and sample size. However, there are a few issues to be addressed for potential audiences of this quality journal. The authors of this study should provide a quality rationale for the following questions:
a. As an experienced statistician, any effect or difference observed in your study would be statistically significant. However, you may not or are not successful to discuss those significant findings (i.e., effects, relationship, or difference). You may provide a more in-depth discussion on your findings
b. Moderating variables in your research are gender and age. Are your findings are new to us? To practitioners, scholars, and coaches in that field?
c. As you commented on p.2, anthropometry, physiology, and training characteristics should be discussed with regard to gender and age, not just gender and age itself to draw meaningful findings
d. I believe that this study is a summary of race participants (finishers) records by age and gender groups. And authors run SPSS to see any possible statistical effect and differences in the dataset. It would be better if you planned every procedural detail ahead of those annual events.
Reviewer 3 Report
The title is applicable.
This paper makes a new contribution in the study area in the following:
- the number of participants was larger than other studies
- exploring differences in the age composition of European and Asian runners
- exploring differences between the proportion of participants in a half marathon and a marathon run compared to Europe in China.
Purpose and objectives of the paper are clearly stated: The aim of this study was to assess the age, gender and the averaged speed characteristics of over 70000 long distance finishers over the last four years.
The study has a sound literature review. Literature references and the discussion of the literature is of a high standard.
Research methodology is executed. Descriptive statistics were used, and between 11 age groups and two events the interactions were calculated using two-way ANOVA, post-hoc with Bonferroni-corrected tests. The analyzes in the study could possibly be supplemented with some statistical analysis. It would be advisable to use linear regression or cluster analysis, the results of which would further enrich the findings.
Results are interpreted correctly and well prescribed.
Reviewer 4 Report
The authors of this study analyzed the running speed of a large number of half - and marathon participants over a repeated period of time.
It is assumed that the speed was calculated from the running time (finisher time) and the running distance. This was not clearly described in the section methods.
The results confirmed numerous observations that women and older people run slower as well. These observations are not new.
Most likely, the speed corresponds to the finisher time. This then would confirm similar results to those of Leyk et al. for example or Knechtle et al..
To summarize, the results, although not new, confirm trendwise the well known findings
in the literature.
As such, the burden is placed on the authors to explain what is really new and different in this study compared to previous studies.
Not mentioned in the paper is the course profile elevation differences and temeprature conditions ,etc. that may determine the speed effects.
With all fig. and tables the abbreviations should be listed again in the legend.
Round 2
Reviewer 1 Report
Thank you for your effort to improve this manuscript. I am pleased with the answers.